# Differential Modulation of Cerebellar Flocculus Unipolar Brush Cells during Vestibular Compensation

**DOI:** 10.3390/biomedicines11051298

**Published:** 2023-04-27

**Authors:** Dan Liu, Jun Wang, Liuqing Zhou, E Tian, Jingyu Chen, Weijia Kong, Yisheng Lu, Sulin Zhang

**Affiliations:** 1Department of Otorhinolaryngology, Union Hospital, Tongji Medical College, Huazhong University of Science and Technology, Wuhan 430022, China; d202081807@hust.edu.cn (D.L.); ent_wangjun@hust.edu.cn (J.W.); 2013xh0823@hust.edu.cn (L.Z.); etian@hust.edu.cn (E.T.); m202175891@hust.edu.cn (J.C.); entwjkong@hust.edu.cn (W.K.); 2Institute of Otorhinolaryngology, Union Hospital, Tongji Medical College, Huazhong University of Science and Technology, Wuhan 430022, China; 3Department of Physiology, School of Basic Medicine, Tongji Medical College, Huazhong University of Science and Technology, Wuhan 430030, China

**Keywords:** vestibular compensation, unipolar brush cells, unilateral labyrinthectomy

## Abstract

Vestibular compensation is a natural behavioral recovery process following unilateral vestibular injury. Understanding the mechanism can considerably enhance vestibular disorder therapy and advance the adult central nervous system functional plasticity study after injury. The cerebellum, particularly the flocculonodular lobe, tightly modulates the vestibular nucleus, the center for vestibular compensation; however, it is still unclear if the flocculus on both sides is involved in vestibular compensation. Here we report that the unipolar brush cells (UBCs) in the flocculus are modulated by unilateral labyrinthectomy (UL). UBCs are excitatory interneurons targeting granule cells to provide feedforward innervation to the Purkinje cells, the primary output neurons in the cerebellum. According to the upregulated or downregulated response to the mossy fiber glutamatergic input, UBC can be classified into ON and OFF forms of UBCs. Furthermore, we discovered that the expression of marker genes of ON and OFF UBCs, *mGluR1α* and *calretinin*, was increased and decreased, respectively, only in ipsilateral flocculus 4–8 h after UL. According to further immunostaining studies, the number of ON and OFF UBCs was not altered during UL, demonstrating that the shift in marker gene expression level in the flocculus was not caused by the transformation of cell types between UBCs and non-UBCs. These findings imply the importance of ipsilateral flocculus UBCs in the acute response of UL, and ON and OFF UBCs may be involved in vestibular compensation in opposite directions.

## 1. Introduction

Vestibular disorders are conditions marked by postural instability, intolerance to head motion, unsteady walking, nystagmus, and vertigo caused by malfunctions in the vestibulocochlear nerve, the central vestibular pathways, and the inner ear [1]. The incidence rate rose with age, affecting 7.4% of people worldwide [2]. Due to the poor quality of life, they produce, vestibular diseases are co-morbid with psychosomatic disorders, and patients are also more likely to fall due to their unbalanced bodies. Though the causes of vestibular disorders are still largely unknown, which poses a significant barrier to developing effective therapeutic strategies, it’s interesting to note that some vestibular syndromes caused by unilateral vestibular lesions gradually improve. This natural recovery process, called vestibular compensation, is a fascinating therapeutic target and a good model for investigating post-lesion neuronal plasticity [3].

Brain imaging studies of unilateral vestibular patients suggest several brain regions are involved in vestibular compensation, including the vestibular nucleus, cerebellum, somatosensory cortex, hippocampus, and thalamus [4]. The primary brain area for vestibular compensation is the vestibular nucleus in the brain stem; acute peripheral vestibular deafferent induces imbalanced excitability between the vestibular nuclei on both sides, which will be partially reversed by endogenous mechanisms of the cells in the vestibular nucleus, including the local circuitry within the vestibular nucleus, and mutual innervation between the bilateral nuclei [5]; and by the exogenous mechanisms that project from other brain regions, including the cerebellum, the most crucial brain region in movement coordination [6,7,8]. The cerebellum has three lobes, the anterior, posterior, and flocculonodular lobes. Despite being the smallest, the flocculonodular lobe is the main lobe connecting to the vestibular nuclei and is involved in balance and spatial orientation [9,10,11,12]. This lobe is composed of two flocculi on both sides of the nodule in the middle, and importantly, the flocculus is essential for the vestibulo-ocular reflex. Impairments in this region lead to nystagmus and vertigo, two main phenotypes after unilateral labyrinthectomy [13,14].

The major input of the cerebellum is the excitatory mossy fibers originating mainly from the cerebral cortex, the vestibular nuclei, and the spinal cord [15]. Mossy fibers synapse with granule cells, which in turn synapse with the inhibitory Purkinje cells, the only output neurons in the cerebellum cortex, which directly project to the vestibular nuclei [16]. Mossy fibers also form synapses and activate granule cells, providing feedforward amplification to the Purkinje cells [17]. The unipolar brush cells (UBCs) are excitatory interneurons targeting granule cells in the cerebellum, which have two distinct types: mGluR1α and calretinin/mGluR2 UBCs [18]. The two kinds of UBCs are long-lasting upregulated and downregulated, mediated by mGluR1α and mGluR2, respectively, in response to a single mossy fiber glutamatergic input received by the brush-like dendrite, giving ON and OFF simultaneous processing for the mossy fiber inputs [19]. Both ON and OFF UBCs in the cerebellum receive inputs from vestibular nuclei; however, only ON UBCs are also directly innervated by the vestibular ganglion neurons, which synapse with hair cells in the inner ear [20].

To investigate the function of the flocculus UBCs in vestibular compensation, first, we observed the mRNA and protein expression of ON and OFF UBC marker genes, *mGluR1α* and *calretinin*, respectively. Within 8 h after UL, only in ipsilateral flocculus did the expression of these marker genes changes; more intriguingly, mGluR1α expression was increased while calretinin level decreased. Further immunostaining showed no change in the number of flocculus mGluR1α- and calretinin-positive neurons after UL, indicating that this expression modulation by UL did not result in cell type transformation. These findings imply the importance of ipsilateral flocculus UBCs in the acute response of UL, and ON and OFF UBCs may be involved in vestibular compensation in opposite directions.

## 2. Materials and Methods

### 2.1. Animals

Male Sprague Dawley (SD) rats aged 8–10 weeks weighing 200–240 g was purchased from the Experimental Animal Research Center of Hubei Province (Hubei, China). All animals were humanely treated following the Guide for the Care and Use of Laboratory Animals (the National Institutes of Health of the United States, Bethesda, MD, USA) [21], and all experiments were conducted in accordance with the standards established by the Institutional Ethical Committee of Tongji Medical College, Huazhong University of Science and Technology. All the animals were group-housed at a constant temperature of 22 ± 1 °C and 65 ± 5% humidity under a 12-h light/dark cycle condition, with ad libitum access to food and water. Six groups of the experimental animals were created at random: sham controls, 4 h, 8 h, one day, three days, and seven days following UL.

### 2.2. Unilateral Labyrinthectomy

UL was performed as previously described [10,12]. Briefly, the same surgeon used an operating microscope to execute the labyrinthectomy on the right side after the rats were anesthetized by intraperitoneal injection of 40 mg/kg sodium pentobarbital. After local anesthesia, a right retroauricular incision was made to expose the auditory canal and the tympanic bulla. Next, 1 mL of xylocaine (with 1:10,000 adrenaline) was used in the wound margin [11]. The tympanic bulla wall was then opened by an otologic drill. Next, the tympanic bulla, malleus, and incus were removed, the stapedial artery was cauterized, the stapes were removed to open the oval window, and a small hole was made in the bony semicircular canal using the otologic drill. Finally, absolute alcohol was injected into the little orifice after the lymph was sucked to ensure that the vestibule was destroyed. The tympanic bulla was opened in the sham operation without destroying the tympanic membranous and ossicles.

### 2.3. Exclusion Criteria

If any of the following symptoms manifested in animals, they were removed from the trial. [22]: (1) a body weight loss of more than 20%, (2) corneal ulceration, which may result from an inadvertent facial nerve lesion, (3) tympanic cavity bleeding, and (4) abnormal behavioral scoring, such as convulsions, paresis or hemiataxia.

### 2.4. Behavioral Assessment

In behavioral tests [5,23,24], vestibular imbalance symptoms, including spontaneous nystagmus, head tilt, postural asymmetry, and tail-hanging, were scored by a blinded assessor.

Spontaneous nystagmus scores ranged from 6 to 10, with 1 point awarded for every 60 beats per minute (bpm). The rats were gently air-puffed over the head if spontaneous nystagmus vanished, and 1 point was given for every 60 bpm for this evoked nystagmus on a scale of 1 to 5.

For the head tilt, the score was assessed spontaneously based on the inclination of the jaw concerning the horizontal. For a 90° angle, the animal resting down on the lesion’s side or barrel-rolling in that direction received 10 points. If the angle was 60 degrees, 7 points; and if the angle was 45 degrees, 5 points.

For postural deficiencies, the scores were as follows: 10 points for spontaneous barrel rolling; 9 points for barrel rolling elicited by a light touch or puff of air; 8 points for the recumbent position on the deafferented side without leg support; 7 points for some ipsilesional leg support; 6 points for moving on one side or using ipsilesional legs for recumbent support; 5 points for moving with bilateral leg support; 4 points for occasionally falling to the ipsilesional side while moving; and 3 points for moving while leaning to the ipsilesional side; 2 points for barely perceptible asymmetry; 1 point for postural asymmetry that is only apparent when picked up. 

Tall-hanging test [25]: Body rotation was evaluated when animals were hoisted off the ground at the base of their tails. 0 to 6 points were given between 0° and 180°; 8 and 10 points were given 180–360° and more than 360°. 

### 2.5. Quantitative Real-Time PCR

Tissues from the flocculus were carefully isolated on ice [26]. RNA was extracted using a RNeasy mini kit (Invitrogen, Carlsbad, CA, USA) following the manufacturer’s instructions. cDNA was reverse transcribed using a PrimeScript RT reagent kit with gDNA Eraser from TaKaRa (Ohtsu, Japan, Code No. RR047A). The RNA and cDNA of each sample were analyzed for concentrations and purity using a GeneQuant pro-RNA Calculator. Quantitative real-time PCR was performed in triplicate using real-time SYBR Green PCR reagents from Vazyme (Nanjing, China) and the 7300 Real-Time PCR System Applied Biosystems (Foster City, CA, USA) to assess the abundance of different transcripts. 

The primers used in the experiment were as follows: *calretinin* (forward: *5′-TGACTTAAATGGAGACGGCAAATT-3′*, reverse: *5′-TAGGTGGTGAGCTGCTGGATATT-3′*), *mGLuR1α* (forward: *5′-CTCCAACACCTTCCTCAACATTT-3′*, reverse: *5′-ATGACACAGACTTGCCGTTAGAA-3′*), *Gapdh* (forward: *5′-GAAGGTCGGTGTGAACGGAT-3′*, reverse: *5′-CCCATTTGATGTTAGCGGGAT-3′*). The relative expression levels were calculated using the 2^−ΔΔCt^ method and reported as fold changes compared to the control group.

### 2.6. Western Blotting

Under a stereomicroscope on ice, tissue samples from the flocculus were extracted and then homogenized in an ice-cold RIPA Lysis Buffer (Beyotime, Nantong, China) with proteinase inhibitors added. The supernatant was then separated from the homogenates by centrifugation at 12,000 rpm for 15 min at 4 °C. Using an Enhanced BCA Protein Assay Kit from Beyotime in Haimen, China), protein concentrations were calculated. Twenty micrograms of each lysate were transferred to polyvinylidene difluoride (PVDF) membranes after being separated on 12% SDS-polyacrylamide gels. The membranes were treated with tris-buffered saline (TBS) containing 0.1% Tween-20 and 5% non-fat milk for 1 h at room temperature, briefly rinsed in TBS, and then incubated with the appropriate dilution of the primary antibodies listed below: anti-mGluR1α (1:1000, BD Pharmingen, Code No: 556389) and anti-calretinin (1:1000, Swant, Switzerland, Code No: 7697). The appropriate horseradish peroxidase (HRP)-conjugated secondary antibody (1:3000, AntGene, Wuhan, China, Anti-mouse Code No: 115-035-003, Anti-Rabbit Code No: 111-035-003) was then added to the membranes after they had been washed to eliminate any excess primary antibody. The membranes were visualized using BeyoECL Plus (Beyotime). Quantitation of the detected bands was performed with the GraphPad Prism 8.0 (Graphpad Software; La Jolla, CA, USA). 

### 2.7. Immunostaining

Rats were deeply anesthetized with sodium pentobarbital and perfused through the ascending aorta with saline, followed by 4% freshly prepared paraformaldehyde (PFA) in 0.12 M phosphate buffer (PB), pH 7.4. The cerebellum was dissected and post-fixed overnight at 4 °C, equilibrated in 20–30% sucrose and embedded in the OCT for cryostat. The thickness of the sections was 20 µm. After blocking for 30 min at room temperature in phosphate-buffered saline (PBS) (pH 7.2) containing 10% normal donkey serum and 0.1% Triton X-100 (PBST), sections were incubated with a combination of the following primary antibodies overnight at 4 °C in PBST: rabbit anti-calretinin (1:1000; Swant, Switzerland, Code No: 7697), mouse anti-glutamate receptor mGluR1α (1:200; BD Pharmingen, Code No: 556389). After washing with PBS 3 times, sections were incubated with a mixture of Alexa Fluor 488 (Donkey Anti-mouse) (1:200; Jackson ImmunoResearch, Code No: 715-545-151) and Alexa Fluor 680 (Donkey Anti-Rabbit) (1:200; Jackson ImmunoResearch, Code No: 711-625-152) for 1 h, then mounted on glass slides with the fluorescent mounting medium.

Immunofluorescence images of double-labeled sections were acquired with a Nikon Confocal Microscope System. The number of fluorescent cells was counted by ImageJ 1.48v (National Institutes of Health, USA). The same exposure time and lighting settings were used for all image acquisitions. A histology expert is unaware of the animals’ position as experimental subjects carried out the histological quantification.

### 2.8. Statistical Analysis

All data were displayed as the mean ± SEM. Data were analyzed by GraphPad Prism 8.0 Software (Graphpad Software; La Jolla, CA, USA). Statistical significance was determined by Student’s *t*-test, one-way ANOVA followed by Tukey’s post hoc test, or two-way ANOVA with Bonferroni post hoc test. A *p* values under 0.05 were considered significant. 

## 3. Results

### 3.1. UL-Induced Behavior Symptoms Gradually Recovered in 7 Days

To evaluate the recovery process after UL, a battery of behavioral tests was assessed 4 h, 8 h, 1, 3, and 7 days after UL (Figure 1). Acute labyrinthine dysfunction such as nystagmus (Figure 1a), head tilting (Figure 1b), postural asymmetry (Figure 1c), and body rotation in the tail-hanging test (Figure 1d) was observed in all rats after UL; however, all symptoms were relieved three days after UL. As expected, the sham-operated rats showed no ocular motor or postural deficits.

### 3.2. UL Increases mGluR1α While Decreasing Calretinin Levels in the Ipsilesional Flocculus

First, the mRNA levels of *mGluR1α* and *calretinin* were measured to see whether UL influences their expression. In the ipsilateral flocculus, the *mGluR1α* mRNA levels were significantly increased compared with the contralateral side and the control group at 4 h after UL (*p* < 0.05) and recovered at 8 h after UL (Figure 2a). While the *calretinin* mRNA level decreased at 4 and 8 h after UL compared with the contralateral side and the control group (*p* < 0.05), which disappeared one day after UL (Figure 2b). No significant mRNA expression change was observed in the contralateral side.

Then the protein expression of these two proteins was evaluated by western blotting. Similar to the mRNA level, the expression of mGluR1α protein was significantly increased in the ipsilesional flocculus, compared with the sham control group at 4 h after UL, while no difference in protein expression more than 8 h after UL (Figure 3a,b and Appendix A). The sham controls and the contra-lesional side did not exhibit any significant difference.

The expression of calretinin protein in the ipsilateral flocculus decreased only 8 h after UL compared with the sham controls. However, no significant difference in protein expression at 4 h after UL (Figure 4a,b). This discrepancy in protein and mRNA expression at 4 h after UL might be due to the lag of protein synthesis relative to mRNA expression.

### 3.3. The Number of ON and OFF UBC Neurons in the Floccules Is Unaffected by UL

Schematic diagram of flocculus in Figure 5c. mGluR1α and calretinin were immunofluorescently labeled in flocculus slices from control and 4 h after UL rats to determine whether the amount of UBCs in the rat cerebellar flocculus is changed by UL (Figure 5a,b). The two cell types did not co-localize, and OFF UBC has a cell body around 1.3 times larger than ON UBC. The major UBC subclass consisting of smaller cells expresses mGluR1α approximately slightly greater than 60%, whereas a minor subclass of slightly larger cells is calretinin, about less than 40% (Figure 5). Importantly, UL did not affect the number of these two cell types (Figure 5d).

## 4. Discussion

The current study proves that the ipsilateral flocculus ON and OFF UBCs are controlled differently in the first 8 h following UL. First, we observed that the UL-induced behavioral symptoms subsided after seven days. Second, we discovered that the expression of the ON and OFF UBCs markers mGluR1α and calretinin changed only during the first 4 to 8 h following UL, indicating that ON and OFF UBCs may be involved in the acute but not the long-term effects of UL. Third, the number of ON and OFF UBCs in the flocculus did not alter 4 h after UL, indicating that mGluR1α- and calretinin-positive neurons increased their expression without recruiting mGluR1α- and calretinin-negative neurons.

Several lines of evidence indicate that the flocculus plays an important role in vestibular compensation. Harashima et al. found that the overexpression of UBC in the cerebellum of Down syndrome mice may affect model rats’ motor and cognitive function by affecting the electrophysiological characteristics of the cerebellar signal pathway [27]. Excision of the vestibular cerebellum (paraflocculus–flocculus, uvula and nodulus) can significantly delay the process of vestibular compensation [28,29]. It has been suggested flocculus is crucial for initiating but not for maintaining the compensation [30]. However, proteomic analysis suggested both sides of flocculus might involve both acute and chronic phases of vestibular compensation [31], however, the mechanism of flocculus in compensation is still largely unknown. UBCs in the flocculus fall into two classes, ON and OFF, based on their response to mossy fiber input, which transform single mossy fiber signals into long-lasting excitation or inhibition, respectively, impact the activity of granule cells and subsequently affect the activity of Purkinje cells [15,16]. When both subtypes receive common sensory input, the ON/OFF distinction in UBCs will allow mossy signals to diverge, setting up distinct processing pathways within the granule cell layer [17]. Here we found ON and OFF UBC in ipsilateral flocculus might be crucial during the acute phase instead of the chronic phase (Figure 6).

mGluR1α is a G-protein coupled receptor mediates glutamatergic transmission transduced into intracellular secondary messengers, particularly calcium signaling [32]. After UL, MF stimulation leads to a large increase of calcium released from intracellular stores within the brush dendrites of the UBCs, which occurs because of prolonged neuron depolarization that may spread to the cell body [33]. In addition, mGluR1α is a molecular “hub” in cerebellar flocculus essential for the modulation of synaptic transmission and neuronal excitability, and synaptic plasticity that underlies motor learning [34]. Importantly, only mGluR1α in ipsilateral flocculus is increased. Several hours after UL, immediate early gene *c-fos* expression is increased in both sides of vestibular nuclei and flocculus, however, predominantly in the ipsilateral side [35], which might be critical for the rebalancing of the vestibular system activity [36]. Similarly, ipsilateral but not contralateral flocculus acute overexpression after UL might mediate the rebalancing of the vestibular system.

Calretinin is a calcium-binding protein regarded as the marker of OFF UBCs. *Calretinin*-knockout mice have impaired motor control, resulting from the lack of calretinin regulation of intracellular calcium concentration [37]. The ipsilateral mossy fiber from the vestibular nucleus is activated after UL, which might reduce the intracellular calcium in OFF UBCs through mGluR2 [38] and subsequently affect the expression of calretinin. To confirm the downregulation of OFF UBCs, c-fos and calretinin co-immunostaining should be conducted to observe whether the activated OFF UBCs are reduced. Current research also indicates that calretinin neurons receive input from upstream glutamatergic neurons, resulting in feedforward inhibition of the downstream inhibitory neurons. This disinhibition circuit may affect neural excitability after UL [39,40].

There were several limitations to this study. In summary, while there is some evidence to support the hypothesis that calretinin and mGluR1α are unique markers for OFF and ON UBCs in the cerebellar cortex, further research is needed to confirm and expand upon our findings. This could include additional experiments to directly examine the expression patterns of these markers in OFF and ON UBCs, as well as investigations into the functional properties and signaling pathways associated with calretinin and mGluR1α expression in these cell types. This could provide a more comprehensive understanding of the cellular and molecular mechanisms underlying cerebellar function.

## 5. Conclusions

In this study, we have demonstrated that UL can upregulate the mGluR1α and downregulate the calretinin in the flocculus in the early stage during vestibular compensation, which indicated that UBC might play essential roles in the cerebello-vestibular pathway during vestibular compensation.

## Figures and Tables

**Figure 1 biomedicines-11-01298-f001:**
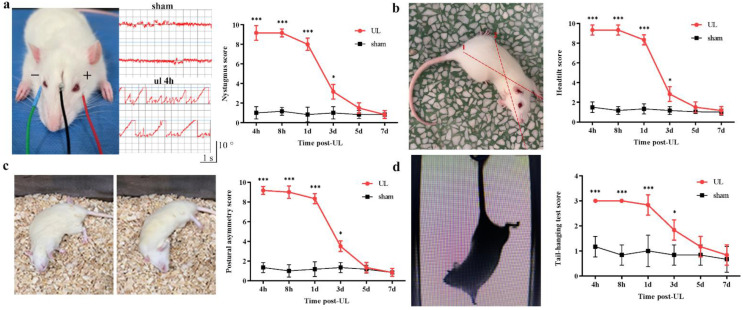
UL-induced behavior symptoms gradually recovered in 7 days. (**a**) After UL, nystagmus was markedly increased but returned to normal after five days. Left: electronystagmography setup and two representative traces of sham and 4 h groups. The positive and negative electrodes were inserted in both orbits’ cavity, and the ground wire was placed on the top of the head. Right: quantitative analysis of all groups (n = 5/group, one-way ANOVA, F (5, 60) = 130.50, * *p* < 0.05, *** *p* < 0.001). (**b**) After UL, head tilt dramatically increased and returned to normal after five days. Left: head tilt was scored according to the angle between the jaw and the horizontal plane. Line 1: The vertical line from the center of the sacral spine to the first thoracic vertebra. Line 2: the tip of the nose to the center of the cranium. Right: quantitative analysis of all groups (n = 5/group, one-way ANOVA, F (5, 60) = 185.30, * *p* < 0.05, *** *p* < 0.001). (**c**) After UL, the Postural asymmetry score also dramatically increased and returned to normal after seven days. Left: different postural asymmetry after UL. Right: (n = 5/group, one-way ANOVA, F (5, 60) = 157.70, * *p* < 0.05, *** *p* < 0.001). (**d**) After UL, the Tail-hanging test score dramatically increased and returned to normal after seven days. Left: schematic diagram of the tail-hanging test. Right: (n = 5/group, one-way ANOVA, F (5, 60) = 12.93, * *p* < 0.05, *** *p* < 0.001).

**Figure 2 biomedicines-11-01298-f002:**
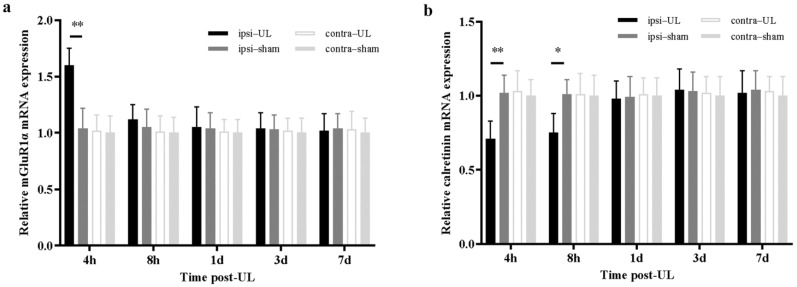
UL increases *mGluR1α* while decreasing *calretinin* mRNA levels in the ipsilateral flocculus. (**a**) Relative *mGluR1α* mRNA expression increased in the ipsilateral flocculus 4 h after UL (n = 5/group, Student’s *t*-test, t = 2.65, ** *p* < 0.01). (**b**) Relative *calretinin* mRNA expression in the ipsilateral flocculus decreased in 4 and 8 h after UL (n = 5/group, Student’s *t*-test, t = 2.34, * *p* < 0.05, ** *p* < 0.01).

**Figure 3 biomedicines-11-01298-f003:**
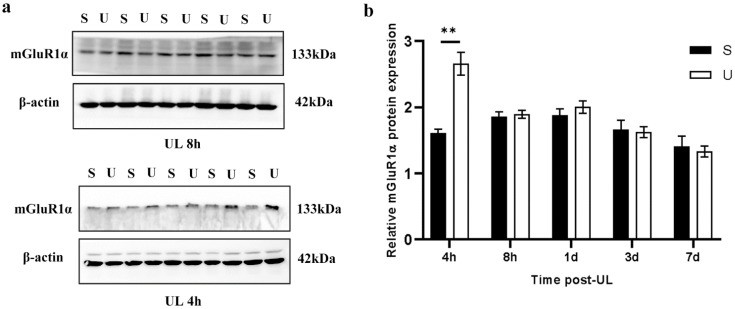
UL Increases mGluR1α Expression in the Ipsilateral Flocculus. (**a**) Western blot for mGluR1α in the ipsilateral flocculus 4 h and 8 h after UL (U) and sham control (S) (n = 5 for each group). (**b**) Quantification of mGluR1α protein expression in the sham (S) and ipsilateral-UL (U) flocculus at 4 h, 8 h, 1 d, 3 d and 7 d after UL (one-way ANOVA, F (4, 40) = 9.32, ** *p* < 0.01).

**Figure 4 biomedicines-11-01298-f004:**
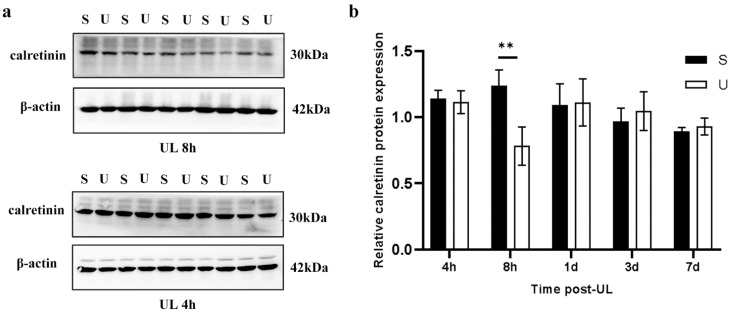
UL decreases calretinin Expression in the Ipsilateral Flocculus. (**a**) Western blot for calretinin in the ipsilateral flocculus at the 4 h and 8 h following UL compared to sham controls (n = 5 for both UL and sham groups). (**b**) Quantification of the calretinin protein expression in the contralateral-sham (S) and ipsilateral-UL (U) flocculus at 4 h, 8 h, 1 d, 3 d and 7 d post-UL (one-way ANOVA, F (4, 32) = 1.91, ** *p* < 0.01).

**Figure 5 biomedicines-11-01298-f005:**
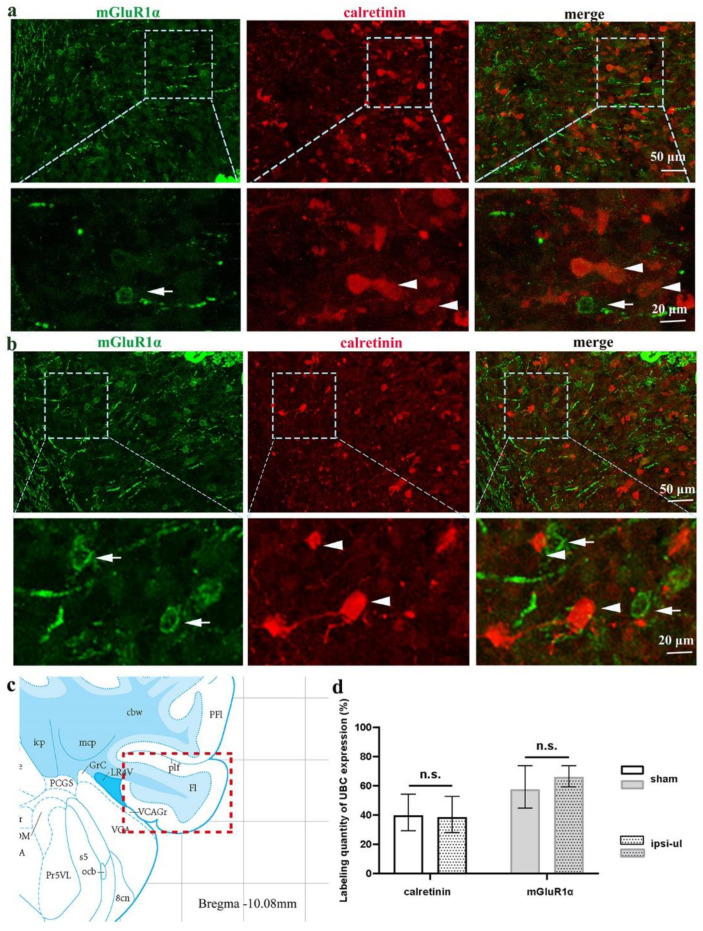
The number of ON and OFF UBC neurons in the floccules is unaffected by UL. Representative images showing mGluR1α- and calretinin-positive neurons in the floccules slices of sham (**a**) and post-UL (**b**) rats. Arrows indicate mGluR1α, and triangles indicate calretinin, respectively. (**c**) Schematic diagram of flocculus. (**d**) Labeling quantity of ON and OFF UBC expression in Flocculus in sham and ipsilateral-UL rats. Data represent mean ± SEM. n.s., no significant difference, followed by Bonferroni’s test, by two-way ANOVA.

**Figure 6 biomedicines-11-01298-f006:**
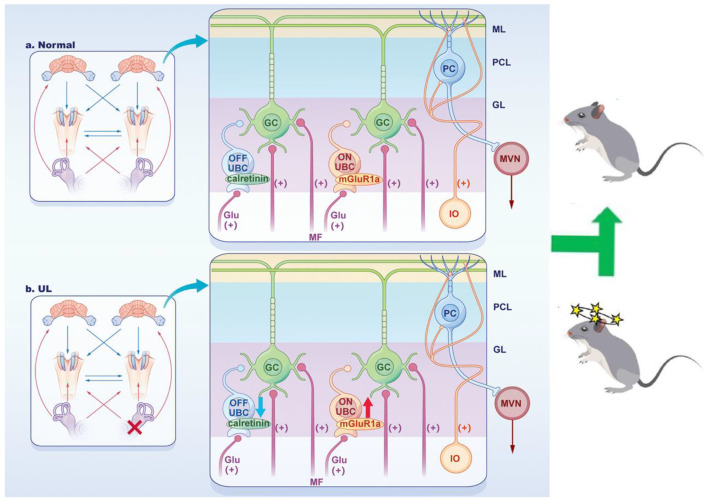
Signal Pathway hypothesis diagram of Cerebellum and Vestibular Cerebellum Pathway. Mossy fibers transmit signals from different regions of the central nervous system to the granular cell layer (GC, UBC, etc.). Granular cells transmit signals to Purkinje cells via parallel fibers. In addition, Purkinje cells also receive climbing fiber excitation input projected from the inferior olivary nucleus (IO). Purkinje cells are the only output neurons in the cerebellar cortex that inhibit MVN in the vestibular cerebellar pathway. (**a**) At normal state, UBC receives a single glutamatergic mossy fiber input. ON UBC has the response of prolonging depolarization and enhancing ignition point to mossy fiber input, mainly mediated by the mGluR1α receptor. Whereas OFF UBC showed prolonged hyperpolarization and reduced ignition point response, mainly mediated by the calretinin receptor. (**b**) Compensation after UL, the afference importation is weakened or lost. Different changes in ON UBC and OFF UBC lead to changes in the level of functional activity of the floccule, further promoting vestibular compensation. The two types of UBC may represent dimensions of maneuverability for floccular in responding to different environmental events. PF, parallel fibers. PC, Purkinje cells. ML, Molecular layer. PCL, Purkinje cell layer. GL, granular layer. GC, granular cell. ON UBC, ON unipolar brush cell. OFF UBC, OFF unipolar brush cell. IO, inferior olive. MVN, medial vestibular nucleus. Glu, glutamic acid. (+) excitatory action; (−) inhibitory action.

## Data Availability

Not applicable.

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
