# Peer review of "Differential Modulation of Cerebellar Flocculus Unipolar Brush Cells during Vestibular Compensation"

_biomedicines, 2023, doi:10.3390/biomedicines11051298_

Round 1

Reviewer 1 Report

In this manuscript from Liu et al., the authors have sought to investigate the role of unipolar brush cells (UBC) in vestibular compensation. In a unilateral labyrinthectomy rat model, the authors observed that mGlur1a and calretinin, markers for ON and OFF UBS respectively were differentially regulated during the first 8 hours after the labyrinthectomy. Overall, the experimental methodology and analyses seem to be fine. The results presented support the authors' conclusions. My comments to the authors are:

Major:

1)      While the behavioral mRNA and protein expression have multiple time points, why is it that only one-time point was looked at for histology? Isn’t it possible that UBC subtypes would have changed at 8 hours, and the authors simply missed the change by looking at only 4 hours? In my opinion, including an 8-hour time point will substantially strengthen the conclusions.

2)      Labeling all the UBCs with eomesodermin (Tbr2) and reporting mGlur1a or calretinin positive UBCs normalized to the total UBC might have detected “recruitment” changes if any, better.  

Minor:

A couple of typos:

Page 2, line 13: nodule instead of nudule

Page 11, discussion, paragraph 2: floccular. Shouldn’t this be flocculus?

Reviewer 2 Report

Manuscript ID:   biomedicines-2247954

Title:     Differential Modulation of Cerebellar Flocculus Unipolar Brush Cells during Vestibular Compensation

Authors:              Dan Liu, Jun Wang, Liuqing Zhou, E Tian, Jing-yu Chen, Wei-jia Kong, Yisheng Lu, Sulin Zhang

General comments:

In this study, the authors addressed neural mechanisms of vestibular compensation following unilateral vestibular injury. It appears to provide some interesting results. Nevertheless, I found some serious concerns as described below.

Major comments:

1.     Page 2, lines 18-19: “Mossy fibers directly synapse to the inhibitory Purkinje cells” The authors appear to lack basic knowledge of the physiology of the cerebellar cortex. Mossy fibers never directly synapse with the Purkinje cells.

2.     The authors assume that Calretinin and mGluR1α are unique markers for OFF UBCs and ON UBCs in the cerebellar cortex. Namely, the authors assume that any other neurons in the cerebellar cortex do not express either Calretinin or mGluR1α. Nevertheless, I was not able to find any reference for the assumption. Therefore, the authors must provide reliable references for it.

3.     It is generally accepted that UBCs are excitatory (glutamatergic) cerebellar granular layer interneurons whose brush-like dendrites receive one-to-one mossy fiber inputs (e.g., Kim et al., Cerebellum 2012, 11(4): 1012–1025). Indeed, the authors stated that “The unipolar brush cells (UBCs) are excitatory interneurons targeting granule cells in the cerebellum (the second paragraph on page 2)”. Nevertheless, in Figure 6 (also in the last sentence on page 10), UBCs are contradictorily described as either glycinergic (most probably inhibitory) or GABAergic (most probably inhibitory). The contradiction is serious.

4.     In this study, animals were anesthetized only by intraperitoneal injection of 40 mg/kg sodium pentobarbital. However, the method may no longer be acceptable for physiological studies because sodium pentobarbital cannot provide enough analgesic effects.

Reviewer 3 Report

The manuscript “Differential Modulation of Cerebellar Flocculus Unipolar Brush Cells during Vestibular Compensation” by Liu et al is a research article which examined whether the flocculus on both sides is involved in vestibular compensation. The authors found that the expression of marker genes of ON and OFF unipolar brush cells (UBCs), mGluR1α and calretinin, was increased and decreased, respectively, only in ipsilateral flocculus following unilateral labyrinthectomy (UL). Interestingly, the number of ON and OFF UBCs was not changed during UL, suggesting that the shift in marker gene ex-pression level in the flocculus was not caused by the transformation of cell types between UBCs and non-UBCs. Thus, the authors concluded that ON and OFF UBCs may be involved in vestibular compensation in opposite directions. Generally, the subject is of interest and scientifically sound and contains essential contents. This paper is also of importance for providing us the important evidence that the flocculus on both sides is involved in vestibular compensation. The manuscript has been well organized and written. However, I have sone concerns on the paper.

1. In this manuscript, statistical significance was assessed by Student’s t-test. The authors should describe t values in the text.

2. In bar graphs, all the data plots should be displayed if possible. The readers can obtain more information from these data.

Round 2

Reviewer 1 Report

The authors have addressed reviewer comments adequately. The revised manuscript is now much improved.

Reviewer 2 Report

I am happy with the revised version. I appreciate your efforts.